# Soil Moisture Regulation under Mulched Drip Irrigation Influences the Soil Salt Distribution and Growth of Cotton in Southern Xinjiang, China

**DOI:** 10.3390/plants12040791

**Published:** 2023-02-09

**Authors:** Pingru He, Jingang Li, Shuang’en Yu, Tao Ma, Jihui Ding, Fucang Zhang, Kaiwen Chen, Shuaishuai Guo, Suhan Peng

**Affiliations:** 1College of Agricultural Science and Engineering, Hohai University, Nanjing 210098, China; 2Key Laboratory of Agricultural Soil and Water Engineering in Arid and Semiarid Areas of Ministry of Education, Northwest A&F University, Yangling 712100, China

**Keywords:** soil moisture regulation, soil water–salt migration, field capacity, mulched drip irrigation, cotton yield

## Abstract

Water deficiency, together with soil salinization, has been seriously restricting sustainable agriculture around the globe for a long time. Optimal soil moisture regulation contributes to the amelioration of soil water and salinity for crops, which is favorable for plant production. A field experiment with five soil water lower limit levels (T1: 85% FC, T2: 75% FC, T3: 65% FC, T4: 55% FC, and T5: 45% FC, where FC is the field capacity) was conducted in southern Xinjiang in 2018 to investigate the responses of soil water–salt dynamics and cotton performance to soil moisture regulation strategies. The results indicated that in the horizontal direction, the farther away the drip irrigation belt, the lower the soil moisture content and the greater the soil salinity. In the vertical direction, the soil moisture and soil salinity increased first and then decreased with an increase in soil depth after irrigation, and the distribution was similar to an ellipse. Moreover, the humid perimeter of soil water and the leaching range of soil salt increased with a decrease in the soil moisture lower limit. Though more soil salt was leached out for the T5 treatment at the flowering stage due to the higher single irrigation amount, soil salinity increased again at the boll setting stage owing to the long irrigation interval. After the cotton was harvested, soil salt accumulated in the 0–100 cm layer and the accumulation amount followed T3 > T5 > T1 > T2 > T4. Moreover, with a decline of soil moisture lower limit, both plant height and nitrogen uptake decreased significantly while the shoot–root ratio increased. Compared with the yield (7233.2 kg·hm^−2^) and water use efficiency (WUE, 1.27 kg·m^−3^) of the T1 treatment, the yield for the T2 treatment only decreased by 1.21%, while the WUE increased by 10.24%. Synthetically, considering the cotton yield, water–nitrogen use efficiency, and soil salt accumulation, the soil moisture lower limit of 75% FC is recommended for cotton cultivation in southern Xinjiang, China.

## 1. Introduction

Water deficiency, together with soil salinity, is not a problem only in China but also a common problem across the globe, particularly in coastal and arid regions. Due to unique climate conditions and wide planting areas, the Xinjiang Uygur autonomous region has become the largest cotton production base in China; its annual cotton yield (5.889 × 10^9^ kg) accounts for 84.9% of the total national output [1]. However, the extremely arid local climate conditions and the unreasonable utilization of water and soil resources over the years have resulted in the secondary salinization of the cultivated land. Freshwater deficiency and soil salinization have become the two key factors that restrict the sustainable development of cotton production in Xinjiang [2,3,4]. Though water-saving irrigation techniques (such as drip irrigation under plastic film) have been applied for crop cultivation in recent years [5,6], improper irrigation strategies may also result in crop yield reduction and soil salinization. Thus, it is necessary to apply effective irrigation strategies for sustainable cotton production with the limited water resources in Xinjiang.

Generally, different from the irrigation strategy triggered by crop evapotranspiration (ETc) [2,7], the soil moisture lower limit is also used as the indicator for triggering irrigation [8,9] since the available soil water in the root zone is in the range of field capacity to wilting coefficient [10,11,12]. According to the principle of “soil salt moves with water”, soil salinity is easily influenced by irrigation regimes, while soil salt distribution follows the soil water flux pattern [13]. According to the point-source infiltration characteristic of drip irrigation, soil salt, along with infiltration water, tends to move toward the fringes of the irrigated area, which then builds a desalinization zone close to the emitters (Figure 1) [14]. Zhang et al. [13] indicated that soil salt mainly accumulated in the surface soil between the adjacent films after crops were harvested, and the amount of soil salt accumulation was 1.24–2.34 times more than that at the depth of 50 cm below the drip tape. Moreover, the soil salt tended to accumulate at the edge of the wetted bulb, and the amount of soil salt accumulation in the surface layer was much greater than that in the deeper layers. Though excessive irrigation can leach the soil salt to a certain extent, heavy irrigation is not conducive to saving water. Thus, it is necessary to investigate the proper irrigation amount for the effective regulation of soil salt. Wang et al. [15] found that drip irrigation with lower soil water lower limits exhibited significant positive impacts on soil salt regulation, while Li et al. [6] revealed that more soil salt moved outside the film with higher soil water lower limits under mulched drip irrigation. Moreover, the increase in irrigation frequency and decrease in irrigation amount for each application were conducive to preventing the upward rise of underground water [16]. It is obvious that the greater the irrigation amount, the deeper the leaching salt depth and the better the leaching effect [7,17,18]; however, the inconsistent irrigation amount and irrigation times regulated by soil moisture increase the difficulty in exploring the distribution of soil water and soil salt.

Many studies have been carried out on the effects of soil moisture lower limits on the plant growth and yield of field crops such as wheat [19,20,21], potato [22], cotton [23], tomato [24,25], and other crops. Generally, a proper increase in irrigation amount contributes to crop growth and yield formation in arid and semiarid areas; however, a large amount of irrigation water does not always correspond to maximum yield and the highest water use efficiency (WUE) [20,25]. Meng et al. (2016) indicated that soil water regulations with moderate deficits (50–60% FC) at each cotton growth stage exhibited significant facilitation effects on cotton root growth, which could effectively regulate the shoot–root ratio [26].

Although many studies have been carried out to determine the proper soil water limit regulation for cotton planting according to crop growth and yield [23,27,28], the distribution of soil water and salt was rarely involved. Moreover, the studies mainly concentrated on cotton cultivation in northern Xinjiang [29,30]. Thus, the experiment was conducted with five soil moisture lower limit levels, which were 85%, 75%, 65%, 55%, and 45% FC, in the Korla region in southern Xinjiang, China. The objectives of this study are (1) to investigate the spatial distribution and temporal variation of soil moisture and soil salt; (2) to evaluate the influence of various soil moisture lower limit regulations on plant growth, cotton yield, water, and nitrogen use efficiency; (3) to optimize a proper irrigation schedule for cotton production with soil water lower limit regulation in southern Xinjiang, China.

## 2. Results

### 2.1. Effects of Soil Moisture Lower Limit Regulation on Soil Water Content during Cotton Growth Period

Generally, the inconsistent soil moisture lower limit regulation strategies resulted in different distributions of soil water content. The variation of soil water content distribution in the 0–100 cm layer at cotton growth stages is shown in Figure 2. Compared with the initial average soil moisture before seeding, which was 20% in the 0–40 cm layer and 26% in the 40–100 cm layer (Table 1), the soil moisture at the budding stage in the 0–40 cm layer (in the range of 13% to 18%) decreased, while no considerable variation of soil moisture for treatments was observed in the 40–100 cm soil layer. Moreover, the soil water content in the 0–20 cm layer inside the film for T1, T2, and T3 treatments was slightly lower than that outside the mulch, which may have resulted from the effective rainfall on June 8th.

At the flowering stage, the irrigation times for treatments were inconsistent due to the differently designed levels of soil moisture lower limits. On 19 July, drip irrigation with the amounts of 32 and 45 mm was applied for the T2 and T3 treatments, respectively, while on 23 July, irrigation water with the amounts of 19, 58, and 70 mm was applied for the T1, T4, and T5 treatments, respectively. According to the soil samples collected on 26 July, the average soil moisture at the flowering stage increased by 8.22%, 2.82%, −5.68%, 13.87%, and 11.81%, respectively, compared with that at the budding stage. Moreover, the lower the soil moisture lower limit, the larger the humid region. Obviously, the soil moisture in the 0–60 cm layer at the cotton flowering stage for the T4 and T5 treatments was significantly higher than that for the T1, T2, and T3 treatments. Though soil water content in the 0–40 cm layer for the T1 treatment was significantly higher than that for the T2 and T3 treatments, soil moisture in the 60–100 cm layer was significantly lower than that for the T2 and T3 treatments.

At the boll setting stage, irrigation water with an amount of 58 mm was applied for the T4 treatment on 9 August, while other treatments were irrigated on 19 August. The soil water content of the soil samples measured on 22 August increased by 2.64%, 2.40%, 11.59%, −11.18%, and 8.30%, respectively, compared with that at the flowering stage. Moreover, except for the T4 treatment, the soil water content in the 0–40 cm and 40–100 cm layers increased with the decline in the soil moisture lower limit.

At the boll opening stage, the corresponding irrigation quotas of the T1, T2, and T4 treatments were 38, 32, and 58 mm, respectively, while T3 and T5 treatments were not irrigated at the boll opening stage, as the soil moisture for them were always higher than the soil moisture lower limit. Irrigation was terminated on 29 August at the boll opening stage; the average soil moisture after harvest decreased by 0.95–13.90% when compared with the boll setting stages, and the difference between treatments was lower than that at the cotton flowering stage and the boll setting stage. Moreover, the higher the soil moisture lower limit, the larger the humid region. For instance, the soil moisture value of 18% corresponds to the 50–60 cm layer for the T3 treatment and the 30–40 cm layer for the T5 treatment, while the soil moisture value of 23% corresponds to the 70–90 cm layer for the T3 treatment and the 50–90 cm layer for the T5 treatment. 

In general, in the horizontal direction, the soil moisture in the 0–40 cm layer tends to decrease with the increasing distance from the drip taps, which follows drip line > narrow row zone > wide row zone > inter-film zone. However, in terms of soil moisture in the 40–100 cm layer, the water content had little difference both inside and outside the film; in particular, the soil water content in the 80–100 cm layer basically remained unchanged (23% approximately). In the vertical direction, with an increase in soil depth, the soil moisture in the 0–60 cm layer increased, while the soil water content in the 60–100 cm layer decreased.

### 2.2. Response of Soil Salt Distribution to Soil Moisture Lower Limit Regulation at Growth Stages

The variation in soil salt in the profile during cotton growth periods is shown in Figure 3. Compared with the initial soil salt content at the seeding stage (average of 0.45 g·kg^−1^), the soil salinity at the budding stage increased within the range of 1.5–3.5 g·kg^−1^. The distribution of soil salt was similar to several elliptic shapes after drip irrigation; in the vertical direction, the soil salt content increased first and then decreased with the increase in soil depth, while in the horizontal direction, the soil salt content had the lowest value under the drip tap. The surface soil salt under the drip tap was leached to the wide row zone and the narrow row zone or even to the inter-film zone under the large irrigation quota. There was no considerable difference in the accumulation and distribution of soil salt among treatments at the cotton budding stage.

Though the soil salt in the flowering stage was leached to a certain extent by drip irrigation compared with the budding stage, the soil salinity after irrigation was still at a higher level (0.9–2.4 g·kg^−1^). The soil salinity for the T4 and T5 treatments was 1 g·kg^−1^ approximately, which declined by 58.3% when compared with other treatments, which indicated that an irrigation amount of 58 mm could leach the soil salinity in the root zone to a lower level. The vertical depth and horizontal distance for soil salt leaching increased with the decrease in soil moisture lower limits. For instance, salts mainly accumulated at the depths of 40–60 and 70–90 cm for the T1 treatment and the T2 treatment, respectively, while the soil salt for the T3 treatment was mainly washed to the 70–100 cm layer and the soil salt for the T4 and T5 treatments was leached out of the 0–100 cm layer, both inside and outside the film.

At the boll setting stage, soil salt leaching occurred for the T1, T2, and T3 treatments, while soil salt accumulation appeared for the T4 and T5 treatments. Compared with the flowering stage, soil salinity at the boll setting stage in the 0–60 cm layer of the T2 and T3 treatments decreased to 0.7 g·kg^−1^ as a result of the frequent irrigation, while the value for T1, T4, and T5 treatments exceed 1.2 g·kg^−1^. There were fewer irrigation times for the T4 and T5 treatments since the soil moisture had, for a long time, been lower than the designed lower limit. The results showed that the soil desalination degree was not only affected by the single irrigation quota but also influenced by the irrigation frequency. Even though the large irrigation quota of the T4 and T5 treatments can thoroughly leach the salt into the deep soil layer at the flowering stage, the soil salt was gathered up again from deep layers at the boll setting stage. Despite the fact that the small irrigation amounts of the T1, T2, and T3 treatments in the flowering stage could not completely leach the soil salt at one time, the soil salt can be leached to a low level with an increase in irrigation times.

The soil salinity at the boll opening stage for the treatments was in the range of 0.6–2.0 g·kg^−1^. In the vertical direction, soil salinity for the T1 and T2 treatments, both inside and outside the film, gradually decreased with the increase in soil depth, and soil salt accumulated in the topsoil, especially in the 0–20 cm layer in inter-film zone. Soil salts in the top layer were leached to the 0–60 cm layer outside the film and the layer of 30–60, 60–80, and 40–70 cm inside the film for the T3, T4, and T5 treatments, respectively. Meanwhile, the soil salt in the 80–100 cm layer for treatments decreased with the increase in soil depth. In general, soil salinity in the horizontal direction for treatments reached the minimum value under the drip line, followed by the narrow row zone and the wide row zone, and had the maximum value in the inter-film zone. In the vertical direction, after drip irrigation, soil salinity inside the membrane first increased and then decreased with the increase in soil depth, with an oval shape formed under the drip tape.

### 2.3. Impacts of Soil Moisture Lower Limit Regulation on Soil Salt Accumulation during the Cotton Growth Period

Changes in soil salinity during the cotton growth period are shown in Figure 4. According to the soil moisture lower limit regulation, drip irrigation was only performed for the T1, T2, and T4 treatments at the boll opening stage. Except for the 0–10 cm soil layer for the T4 and T5 treatments, soil salt accumulated for the treatments, both inside and outside the film. Soil salt for the T1 and T2 treatments mainly accumulated in the 0–60 cm layer, both inside and outside the film, while the soil salt for the T3 and T4 treatments chiefly accumulated in the 0–80 cm layer inside the mulch and the 0–60 cm soil layer outside the film, which indicated that the depth of soil salt accumulation increased with a decrease in the soil moisture lower limit. There was no significant difference in soil salt accumulation between the T1 and T2 treatments inside the film; however, the accumulation amount of soil salt outside the film for the T1 treatment was significantly higher than that for the T2 treatment. This indicated that the higher the soil moisture lower limit, the smaller the irrigation water’s vertical and horizontal movement range and distance under mulched drip irrigation. Moreover, with the increase in irrigation frequency for the T1 treatment, the wetting peak tended to expand, and more salt migrated outside the film. Due to the larger irrigation quota for the T5 treatment than that for the T3 treatment, the amount of soil salt accumulation for the T5 treatment was less than that for the T3 treatment. Soil salt accumulation in the 0–40 cm layer inside the film for the T5 treatment was not significantly different from that of the T1 and T2 treatments, but outside the film, it was relatively more serious than the T1 and T2 treatments. After drip irrigation for the T4 treatment was performed during the boll opening period, soil salt in the 0–40 cm layer inside the film was leached to the 40–100 cm layer inside the film and the 0–40 cm layer outside the film. Therefore, soil salt in the 0–40 cm layer inside the film for the T4 treatment was significantly lower than that for other treatments. There was no significant difference in the amount of soil salt accumulation in the 0–100 cm layer between the T4 and T2 treatments. In general, owing to the topsoil evaporation outside the film and the soil moisture horizontal migration inside the mulch, the amount of soil salt accumulation inside the film was less than that outside the mulch. Affected by the topsoil evaporation, the amount of soil salt accumulation in the 0–60 cm layer was greater than that in the 60–100 cm layer.

### 2.4. Influence of Soil Moisture Lower Limit Regulation on Cotton Growth, Yield, Water, and Nitrogen Use Efficiency and Correlation Analysis between the Indexes

Drip irrigation based on soil moisture lower limits exhibited a significant influence on plant growth and cotton yield. According to Figure 5, the plant height, LAI, and above-ground dry matter (GDM) of cotton increased with an increase in soil moisture lower limit. Compared with the plant height (84.07 cm), LAI (5.00), and GDM (2.68 × 10^4^ kg·hm^−2^) of the T1 treatment, the plant height for the T2, T3, T4, and T5 treatments decreased by 3.89%, 6.92%, 18.54%, and 34.88%, respectively, the LAI for the T2, T3, T4, and T5 treatments decreased by 16.28%, 21.95%, 38.89%, and 61.29%, respectively, and the GDM for the T2, T3, T4, and T5 treatments decreased by 1.74%, 11.60%, 19.21%, and 23.83%, respectively.

The sufficient soil moisture in the cotton root zone for the T1 treatment corresponded to the high values of plant height and LAI. Meanwhile, the cotton growth was inhibited by soil water stress for a long time for the T5 treatment. The above results indicated that increasing the soil moisture lower limit was beneficial to plant growth. In addition, the plant nitrogen uptake and cotton yield increased gradually with the increase in the soil moisture lower limit (Figure 5). The plant nitrogen uptake and seed cotton yield for the T1 treatment were 262.5 and 7233.2 kg·hm^−2^, respectively. Compared with the T1 treatment, the nitrogen uptake for the T2, T3, T4, and T5 treatments decreased by 3.45%, 18.67%, 25.46%, and 31.96%, respectively, while the seed cotton yield decreased by 1.21%, 9.97%, 18.05%, and 20.78%, respectively. Moreover, the shoot–root ratio increased with the decrease in the soil moisture lower limit, which may be due to the fact that when the cotton was subjected to water stress, the cotton roots were more vigorous in order to absorb the deep soil water. Moreover, both the WUE and NUE increased with the decrease in the soil moisture lower limit and had the highest values in the T5 treatment, with a WUE of 1.63 kg·m^−3^ and an NUE of 33.60 kg·kg^−1^. Compared with the T5 treatment, the WUE of the T1, T2, T3, and T4 treatments decreased by 28.39%, 16.48%, 8.28%, and 9.92% respectively, and the NUE of the T1, T2, T3, and T4 treatments decreased by 21.91%, 19.04%, 8.82%, and 7.28% respectively. 

The correlation among indexes under different soil moisture regulations is shown in Figure 6. Except for soil salt accumulation, there were significant correlations between indexes; positive correlations were found between WUE, the shoot–root ratio, and NUE. The correlation index between cotton dry matter and nitrogen uptake obtained the largest value, which was 0.97. Cotton yield was positively related to plant height, LAI, dry matter, and nitrogen uptake, while it was negatively related to the root–shoot ratio and NUE. The correlation between cotton yield and indexes was as follows: nitrogen uptake (0.74) > dry matter accumulation (0.69) > plant height (0.65) > shoot–root ratio (0.65) > LAI (0.59). This implied that the cotton yield could be evaluated according to cotton plant growth indicators such as plant height, LAI, shoot–root ratio, dry matter accumulation, and plant nitrogen uptake. There was no obvious correlation between the salt accumulation and other indexes, which may be attributed to the fact that the amount of soil salt accumulation was mainly affected by the last irrigation event.

In summary, reducing the lower limit of soil moisture tended to result in an increase in WUE and NUE; however, cotton growth was inhibited and the yield declined. Meanwhile, increasing the lower limit of soil moisture had little difference in yield improvement, while the WUE and NUE were significantly decreased (Figure 7). Thus, with the aim of high yield, high water–nitrogen use efficiency, low soil salt accumulation in cotton planting, and a soil moisture lower limit of 75% FC were appropriate for cotton cultivation in this study.

## 3. Discussion

### 3.1. Spatial and Temporal Distribution of Soil Moisture and Soil Salinity under Mulched Drip Irrigation

Generally, variations in soil moisture and soil salinity, along with the cotton growth period, are susceptible to being affected by irrigation systems. In this study, from the seeding stage to the budding stage, soil moisture decreased gradually as a result of topsoil evaporation. Soil water content increased gradually after drip irrigation at the budding stage and reached the maximum value at the boll setting stage with continuous irrigation but decreased gradually after terminating irrigation at the boll opening stage. Compared with the soil salinity before sowing, the soil salt content increased at the seeding stage but decreased at the budding stage and the boll setting stage. Moreover, soil salinity tended to increase again after irrigation was stopped at the boll opening stage, which was inconsistent with the research of Zhang [5] et al.

Studies have found that after drip irrigation with fresh water, the soil water content under the tape was higher than in other positions, and the corresponding soil salt content was lower [13,31]. In this study, the salt of the topsoil inside the film migrated with the wetting peak after drip irrigation. In the horizontal direction, the farther away the drip irrigation belt, the higher the soil salinity, which followed drip line < narrow row zone < wide row zone < inter film zone. However, the distribution of soil moisture was inverse. In the vertical direction, the soil moisture and soil salinity, both inside and outside the film, first increased and then decreased with the increase in soil depth after irrigation, and the distribution of soil water content and soil salt content was similar to an oval shape. Moreover, the soil salinity inside the film was significantly lower than that outside the mulch in the 0–60 cm layer; however, little difference was found in the 60–100 cm layer, both inside and outside the film, which is consistent with the research [7,32,33] that soil salt accumulates especially in the surface 0–20 cm soil layer but not in the deep 60–80 cm soil layer. Moreover, the soil water humid perimeter contains the vertical depth, and the horizontal distance of soil salt leaching increases with the decrease in the soil moisture lower limit, which is in agreement with the study of Hou et al. [7].

### 3.2. Soil Salt Accumulation with Soil Moisture Lower Limit Regulation

Many studies have proposed that in the vertical direction, soil salt mainly accumulates in the surface layers rather than the deep layers [34,35,36]. The present study found that soil salt accumulated both inside and outside the mulch for treatments at the end of the cotton growth period; soil salt mainly accumulated in the 0–60 cm layer, while little difference in soil salt accumulation was found in the 80–100 cm layer for treatments. Moreover, the soil salt was continuously concentrated in the surface layer because of strong topsoil evaporation and the high groundwater buried depth. According to the study of Liu et al. [37], the difference in soil salinity spatial distribution under mulched drip irrigation only occurred in the crop growth period; the soil salinity was redistributed to a uniform state after stubble tillage. Therefore, proper flood irrigation is necessary for soil salt desalination during the crop’s non-growth period [5,37,38].

In cotton fields, the soil moisture regulation under mulched drip irrigation directly influences the irrigation quota and irrigation times; additionally, increasing irrigation quota and irrigation times is conducive to alleviating soil salt accumulation [39]. Wang et al. [15] found that the leaching effect of high irrigation lower limits on salt is lower than that of low irrigation lower limits as the horizontal distance and vertical scope of the soil wetted volume are smaller. However, the present experiment found that with the decrease in the soil moisture lower limit, although the irrigation wetting area and the salt leaching range were extended, along with the increase in the single irrigation quota, with the decline in irrigation frequency and the total irrigation amount, the value of soil salt accumulation after harvest increased. This is coherent with the result of Li et al. [40], who found that the higher the irrigation lower limit, the higher the soil moisture and the lower the soil salinity in the root zone.

High irrigation frequency could effectively prevent salt accumulation again, and the soil salt leaching efficiency with high irrigation frequency was higher than the low irrigation frequency [41]. In this study, when the soil moisture lower limit was settled at 45% FC, it may have led more soil salt to be leached out of the crop root zone due to the higher single irrigation amount [42], but the irrigation times were relatively few as the soil water content in the planned wetting soil zone was difficult to reach with the designed value. It was easy for cotton roots to absorb soil water from deep soil layers because of the shallow groundwater buried depth of 1.52 m [43]; thus, soil salt accumulated in the surface layer again at the boll opening stage.

### 3.3. Crop Growth, Yield, Water, and Nitrogen Use Efficiency with Soil Moisture Lower Limit Regulation

Crop biomass accumulation and plant height are important indexes to measure the cotton’s growth status. They generally respond to the soil water deficiency first and then are exhibited in the crop yield [44,45]. Liu et al. [46] found that the crop’s vegetative growth was limited to a certain extent by deficit irrigation, which was manifested in the decline of plant height and a reduction of biological yield. Similarly, in this experiment, drip irrigation triggered by too-high soil moisture lower limits (85% FC) tends to result in the overgrowth of cotton plants, while too-low soil moisture lower limits (45% FC) usually result in the undergrowth of cotton plants, both of which were detrimental to high cotton yields. Meanwhile, the long duration of high LAI, a large leaf area of photosynthesis, and slow leaf senescence are an important physiological basis for the high yield of cotton in Xinjiang [47]. This study has shown that the LAI decreased with the reduction in the soil moisture lower limit; for instance, the LAI for the T5 treatment was significantly lower than other treatments due to the cotton plant being subjected to drought stress since the irrigation quota for the T5 treatment was lower than other treatments. This is consistent with the research of Wang et al. [14], who have noted that cotton plant height and LAI decreased with a decrease in the soil water lower limit, as a low irrigation lower limit would produce salt stress and inhibit plant growth.

Additionally, though the shoot growth was more sensitive to soil salt stress than the root growth, the ability of soil water and nutrient absorption by crops were generally affected by plant root growth [48,49]. The present experiment found that the root–shoot ratio increased gradually with a decrease in the soil moisture lower limit. This result was consistent with the research of Wu et al. [50], who found that the soil water stress was conducive to the elongation downward of the roots, which may attribute to the fact that the plant roots adjust themselves in unfavorable soil water environments to achieve a balance in important physiological mechanisms. Studies have concluded that drip irrigation with an irrigation quota of less than 300 mm can obtain a higher WUE but with a low crop yield, while excessive drip irrigation has no obvious effect on increasing yield [39]. In this study, compared with the T1 treatment, the WUE and NUE for the T2 treatment were significantly improved, but the yield only decreased by 1.21%, which was similar to the research of Zhou et al. [51], who found that the soil moisture lower limit of 65–75% FC resulted in the yield being decreased by 7.5%, while the IWUE increased by 11.2% when compared with the soil moisture lower limit of 75–85% FC.

## 4. Materials and Methods

### 4.1. Experimental Site

The field experiment was conducted in Korla, which is a typical salt-alkali region in southern Xinjiang, China (40°53′03″ N, 86°56′58″ E, 900 m above sea level). The region enjoys a typical continental desert climate, with an annual mean precipitation of 56 mm and an annual average potential evaporation of 2417 mm. The annual average sunshine duration for the experimental site is 2941.8 h; moreover, the annual average temperature is 11 °C, and the diurnal amplitude range is 14–15 °C. The total rainfall during the cotton growth period was 18.6 mm in 2018, while the effective rainfall was only 5.6 mm (Figure 8). Moreover, the water surface evaporation initially increased from April to June and then decreased from June to September. Moreover, the average buried depth of the groundwater was 1.53 m (Figure 9).

The soil in the 0–60 cm layer was mainly composed of sandy loam and silt loam, while the 60–100 cm soil was basically sandy (Table 1). The average bulk density in the 0–60 cm soil layer was 1.57 g·cm^−3^, while the average soil FC (volumetric water content) and wilting point (volumetric water content) were 30.57% and 17.32%, respectively. Due to supplementary irrigation in the winter of the previous year, the average initial soil salinity in the 0–100 cm layer was only 0.52 g·kg^−1^, and the content of NO_3_^-^-N and NH_4_^+^-N in the 0–100 cm soil layer was 13.46 and 7.89 mg·kg^−1^, respectively.

### 4.2. Experimental Design and Arrangement

With the root zone soil field capacity acting as the soil water upper limit, the mulched drip irrigation was triggered by five soil water lower limit levels in the designed wetting layer, which were 85%, 75%, 65%, 55%, and 45% FC, respectively. Five treatments were replicated three times in a randomized complete factorial block design with 15 plots.

A large amount (300 mm) of ground irrigation with fresh water was applied on 8 December 2017 to leach the accumulated topsoil salt. Raised beds, 1.06 m in width and 10 m in length, were prepared with a spacing of 0.46 m; the white plastic polyethylene film (1.2 m wide × 10 m long × 0.038 mm thick) was covered on the beds. Meanwhile, cotton of the Xinluzhong 66 variety was sown at a depth of 0.03 m, with a row spacing of 0.10 m + 0.66 m + 0.10 m (Figure 10) and a plant spacing of 0.10 m on 11 April 2018. Each treatment with three plots contained twelve raised beds and was equipped with an independent drip irrigation system; drip taps with 0.3 m emitter intervals and a flow rate of 2.4 L·h^−1^ were placed on the beds under the plastic mulch (Figure 10). From the budding stage to the boll opening stage of cotton, irrigation water with an average salinity of 0.7 g·L^−1^ was applied immediately for each plot as soon as the soil moisture lower limit reached the designed value in the plan wetting layer.

Urea (N ≥ 46%), ammonium dihydrogen phosphate (P_2_O_5_ ≥ 46%), and potassium chloride (K_2_O ≥ 62%) were employed as fertilizers, and N-P_2_O_5_-K_2_O of 300–120–60 kg·hm^−2^ was applied according to the local cotton fertilization practice. Moreover, the fertilizer mass proportion at the budding, flowering, boll setting, and boll opening stages of cotton was 25%, 30%, 30%, and 15%, respectively. Other agronomy practices were the same as the local conventional cotton cultivation practices.

### 4.3. Measurements and Methods

#### 4.3.1. Irrigation Schedule

FC was regarded as the upper limit of soil moisture, and the quota for each drip irrigation event was determined by the following equation:(1)$$M=10\times \gamma \times H\times p\times \left(\theta_{\max}-\theta_{\min}\right)$$
where *M* is the individual irrigation quota (mm) under drip irrigation; *γ* is the soil bulk density of the designed wetting layer (g·cm^−3^); *H* is the designed wetting depth (cm), which was 40 cm at the cotton budding stage, 60 cm from the flowering stage to the boll opening stage; *p* is the moisture ratio, which was 0.7 under drip irrigation; *θ*_max_ and *θ*_min_ are the upper and lower limits of soil moisture (mass moisture content) in the designed wetting layers, respectively.

The crop water consumption (ET) was estimated by the following water balance equation:(2)$$ET=P+U+I-R-D-10\sum_{i=1}^{n}\left[\gamma_{i}H_{i}\left(\theta_{i1}-\theta_{i2}\right)\right]$$
where *ET*, *P*, and *U* represent the amount of crop water consumption, precipitation, and groundwater recharge, respectively (mm); *I*, *R*, and *D* represent the amount of irrigation, surface runoff, and deep leakage, respectively (mm); *i* is the number of soil layers; *n* is the total number of soil layers; *γ_i_* is the soil dry bulk density of layer *i* (g·cm^−3^); *H_i_* is the thickness of soil layer *i* (cm); *θ_i_*_1_ and *θ_i_*_2_ are the soil moisture contents at the beginning and end of the calculation interval, respectively (g·g^−1^). Due to the fact that there was no surface runoff and deep leakage in the experimental area because of the limited drip irrigation and precipitation, *R* = 0, *D* = 0.

The amount of groundwater recharge during the cotton growth period was determined by a formula with groundwater depth and water surface evaporation, as proposed by Averiyanov [52]: (3)$$U=\partial \times (1-{\left(\frac{h}{3.8}\right)}^{3})\times E_{0}$$
where *∂* represents the coefficient of groundwater recharge, which is 0.6 here; h represents the groundwater buried depth (mm); *E*_0_ represents the water surface evaporation (mm).

The detailed irrigation schedule is listed in Table 2. During the crop growth period, the irrigation times for the T1, T2, T3, T4, and T5 treatments were 22, 12, 7, 5, and 4 times, respectively. The total irrigation amount ranged from 201 to 378 mm, and the crop water consumption was in the range of 367–568 mm. Obviously, the total irrigation quota, together with the crop water consumption, increased with the increase in soil moisture lower limit.

#### 4.3.2. Plant Height, Above-ground Dry Matter, Nitrogen Uptake, and Root–Shoot Ratio 

At the boll opening stage of cotton, six representative plants were randomly selected and dug out from the soil. A ruler was used to measure the plant height, and scissors were used to separate the plant samples into roots, stems, leaves, and bolls. All the plant samples were dried in an oven at 105 °C for 30 min and then at 75 °C to a constant weight to determine the biomass of each cotton organ. Afterwards, the plant samples of different organs were milled and then screened through a 0.5 mm sieve; the nitrogen concentration was determined with the micro-Kjeldahl method [53]. The nitrogen uptake was calculated as the product of nitrogen concentration and dry weight for each plant tissue.

#### 4.3.3. Leaf Area Index

The specific leaf area (*SLA*) method was used to measure the leaf area of cotton [33], and the *LAI* was calculated as follows:(4)LAI=SLA/SA
(5)$$SA=\frac{Gr\times W_{f}\times D_{h}}{Rn}$$
where *SLA* is the leaf area of a single plant (cm^2^); *SA* is the land area occupied by a single plant (cm^2^). Gr is the germination rate; *W_f_* is the width of the plastic film (*W_f_* = 1.52 m); *D_h_* is the distance between hills (*D_h_
*= 0.1 m); *Rn* is the row number (*Rn* = 4).

#### 4.3.4. Cotton Yield

Cotton was harvested by hand on 17 September 2018. Six blocks (1.52 × 1.0 m) were selected randomly in each plot for yield measurement, and the cotton yield was weighted by an electronic balance.

#### 4.3.5. Nitrogen Use Efficiency and Water Use Efficiency

Nitrogen use efficiency (*NUE*, kg·kg^−1^) and *WUE* (kg·m^−3^) were determined as follow:(6)NUE=Y/TNU
(7)WUE=Y/ET
where *Y* denotes the cotton yield (kg·hm^−2^); *TNU* denotes the total nitrogen uptake of cotton (kg·hm^−2^).

#### 4.3.6. Soil Water Content and Salinity

Soil samples for moisture determination were collected at position B in Figure 10 every day, with an auger (5 cm in diameter and 25 cm in height) during the growth period of cotton at a depth of 0–20, 20–40, and 40–60 cm, respectively. Additional soil samples for soil water and soil salinity measurement were taken at positions A (wide row zone, abbreviated as WRZ), B (drip line, abbreviated as DI), C (narrow row zone, abbreviated as NRZ), and D (inter-film zone, abbreviated as IFZ) at the soil depth of 0–10, 10–20, 20–40, 40–60, 60–80, and 80–100 cm at the end of each cotton growth stage (budding, flowering, boll setting, and boll opening stages).

Soil samples for soil moisture measurement were sealed, weighed, dried in a fan-assisted oven at 105 ± 2 °C for 24 h, and reweighed to determine the gravimetric water content. The soil samples for salinity measurement were air-dried, ground, passed through a 1 mm sieve, and then mixed with distilled water to a mass ratio of 1:5. Electrical conductivity (EC1:5) was measured by a conductivity meter (DDS-307, Shanghai Precision & Scientific Instrument Inc., Shanghai, China). The soil salt content (SC) was estimated according to a linear statistical relationship (SC = 2.446 × EC_1:5_; R^2^ = 0.98).

#### 4.3.7. Soil Salt Accumulation

The total soil salt accumulation could be described by the soil salt accumulation inside and outside the film based on the weights of horizontal distance to the sampling positions. The salt accumulation inside the plastic film can be calculated by the salt accumulation in the wide row zone (point A in Figure 10), drip line (point B in Figure 10), and narrow row zone (point C in Figure 10) in different soil layers. Additionally, the variation of soil salinity in inter-film zone (point D in Figure 10) during the cotton growth period represents the soil salt accumulation outside the film [54]. Soil salt accumulation during the whole growth period of cotton can be evaluated as follows:(8)$$\Delta S_{in}{}_{,i}=10H_{i}\gamma_{i}\left(\frac{23}{106}\Delta SC_{i,A}+\frac{38}{106}\Delta SC_{i,B}+\frac{45}{106}\Delta SC_{i,C}\right)$$
(9)$$\Delta S_{out}{}_{,i}=10H_{i}\gamma_{i}\Delta SC_{i,D}$$
(10)$$\Delta S_{i}=\frac{106}{152}\Delta SC_{in}{}_{,i}+\frac{46}{152}\Delta SC_{out}{}_{,i}$$
where Δ*S_i_* represents the soil salt accumulation amount in layer *i* (*i* = 1, 2, ..., 6) (g·m^−2^); ∆*S_in,i_* and ∆*S_out,i_* represent the soil salt accumulation quantity in layer *i* inside and outside the film, respectively (g·m^−2^); Δ*SC_i,A_*, Δ*SC_i,B_*, Δ*SC_i,C_*, and Δ*SC_i,D_* are the variations in soil salinity for points A, B, C, and D from sowing to harvest in layer *i*, respectively (g·kg^−1^); *H_i_* is the soil thickness of layer *i* (m); *γ_i_* is the soil bulk density of layer *i*.

#### 4.3.8. Observation of Meteorological Data and Groundwater Depth

The meteorological data, including daily temperature, rainfall, and wind speed, were obtained from an automatic weather station (YM-03A) that was 50 m away from the experimental field area (Figure 10). Daily water surface evaporation was measured every day at 18:00 with an evaporation pan. The groundwater buried depth in the field during the cotton growth period was observed by an automatic record water level meter.

### 4.4. Data Analysis

One-way analysis of variance (ANOVA) was performed with Duncan’s multiple range test at *p* < 0.05 in SPSS 20.0 (SPSS Inc., Chicago, IL, USA) to evaluate the influence of soil moisture regulation on plant height, dry matter, LAI, shoot–root ratio, yield, WUE, NUE, and soil salt accumulation. The effects were analyzed for average and standard deviation for each treatment (*n* = 6). The distribution of soil water content and soil salinity were analyzed by Surfer 12.0. Moreover, software such as Origin 9.0, RStudio 4.2.0, and Auto CAD 2016 were used to produce the figures.

## 5. Conclusions

A higher level of the soil moisture lower limit is conducive to maintaining a suitable root zone condition, with proper soil water content and less soil salt accumulation for cotton plants’ vigorous growth and high yield. After drip irrigation, the topsoil salt moved with the wetting peak to the deep soil layers and on both sides of the drip tape; with the increase in distance, apart from the drip tape, the soil salinity increased gradually and followed drip line < narrow row zone < wide row zone < inter-film zone, while the soil water content increased. In the vertical direction, the soil moisture and soil salinity for each layer after irrigation first increased and then decreased with the increase in soil depth, and the distribution was similar to oval shapes. Moreover, the soil water humid perimeter and the range of soil salt leaching increased with the decline in the soil moisture lower limit. Though more soil salt was leached out for the T4 and T5 treatments at the flowering stage, soil salinity increased again at the boll setting stage. Similarly, although the soil salt for the T1 and T2 treatments could not be completely washed out at the flowering stage, the soil salinity decreased at the boll setting stage with the increase in irrigation times. Moreover, with the decline in the soil moisture lower limit, the plant height, LAI, GDM, plant nitrogen uptake, and cotton yield decreased significantly while the shoot–root ratio, WUE, and NUE increased. Synthetically, considering the cotton yield, WUE, NUE, and soil salt accumulation, the soil moisture lower limit at 75% FC is appropriate for cotton cultivation in southern Xinjiang, China. The irrigation schedule should have a ground irrigation amount of 300 mm during the non-growing season and a mulched drip irrigation quota of 334 mm (12 irrigation times) during the growth period of cotton.

## Figures and Tables

**Figure 1 plants-12-00791-f001:**
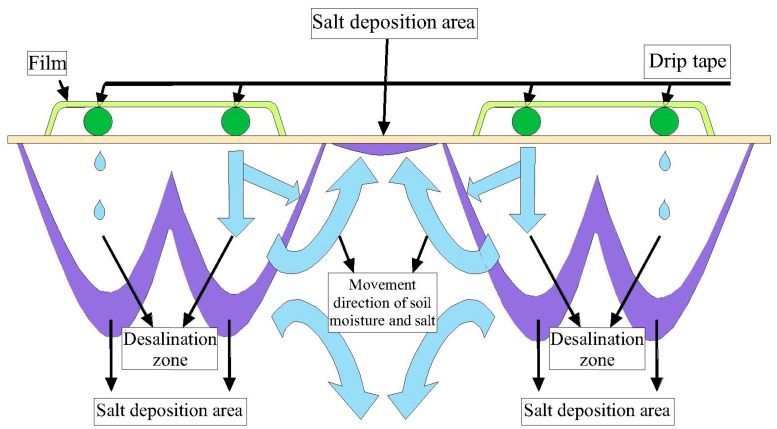
Soil salt transport under mulched drip irrigation.

**Figure 2 plants-12-00791-f002:**
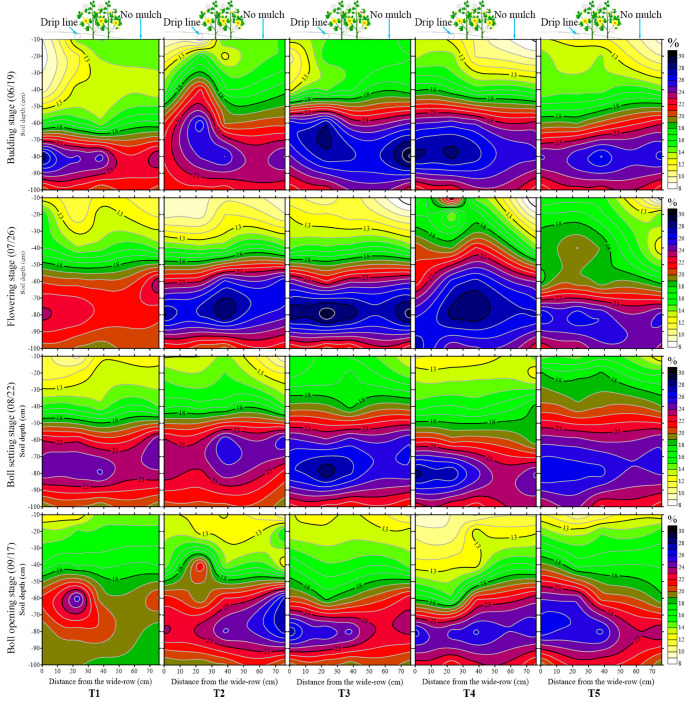
Dynamics of soil moisture during the cotton growth period.

**Figure 3 plants-12-00791-f003:**
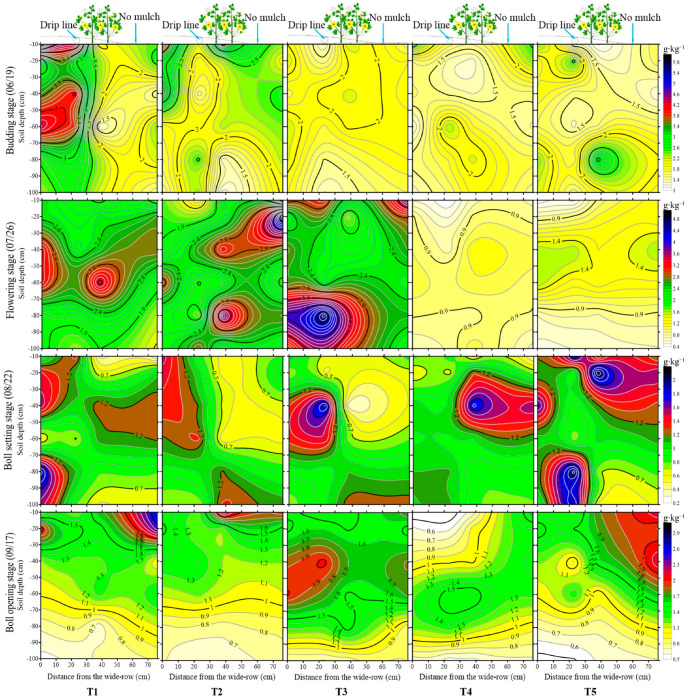
Variation of soil salinity during the cotton growth period.

**Figure 4 plants-12-00791-f004:**
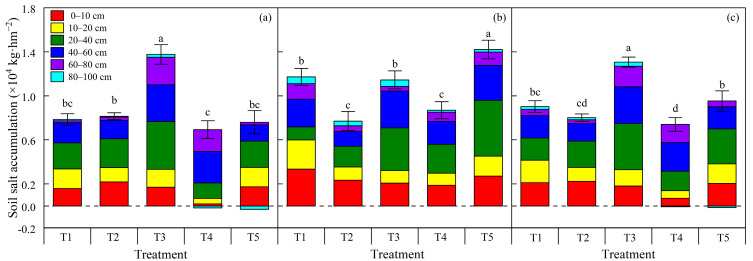
Soil salt accumulation at the end of the cotton’s growth period. (**a**) Soil salt accumulation inside the film; (**b**) soil salt accumulation outside the film; (**c**) the total soil salt accumulation. Bars are the one standard error of the mean (n = 3). Different letters above the error bars indicate a significant difference at *p* < 0.05 according to the Duncan test.

**Figure 5 plants-12-00791-f005:**
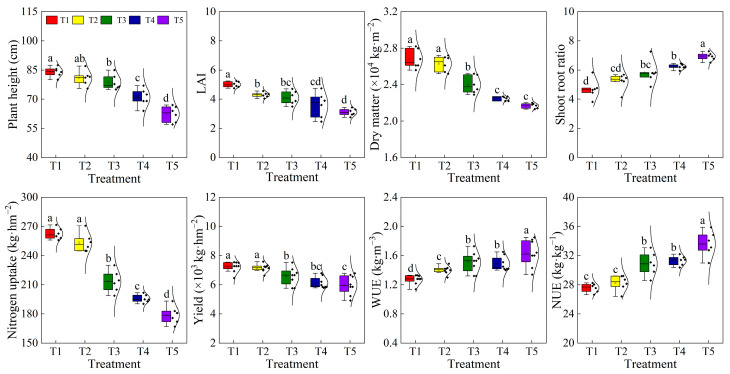
Plant height, LAI, dry matter, shoot–root ratio, nitrogen uptake, yield, WUE, and NUE of cotton at the boll opening stage. Different letters above the boxes indicate a significant difference at *p* < 0.05, according to the Duncan test.

**Figure 6 plants-12-00791-f006:**
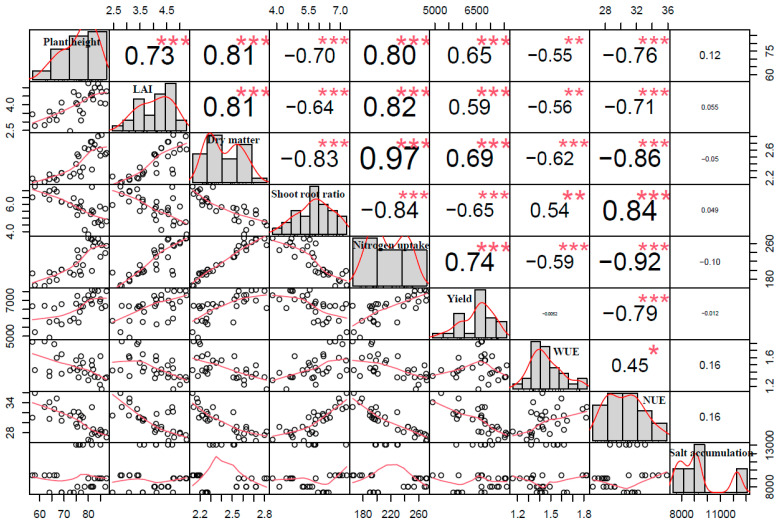
Multi-panel scatter plots of tested indexes, including the plant height, LAI, dry matter, root–shoot ratio, nitrogen uptake, yield, WUE, NUE, and salt accumulation of cotton. Phenotypic traits with their histograms are given in the diagonal panel. Lower panels represent pairwise scatter plots, with red lines depicting the best fit, and the upper right panels show Pearson correlation coefficients. Symbols “***”, “**”, and “*” indicate the significance levels of *p* < 0.001, 0.01, and 0.05, respectively.

**Figure 7 plants-12-00791-f007:**
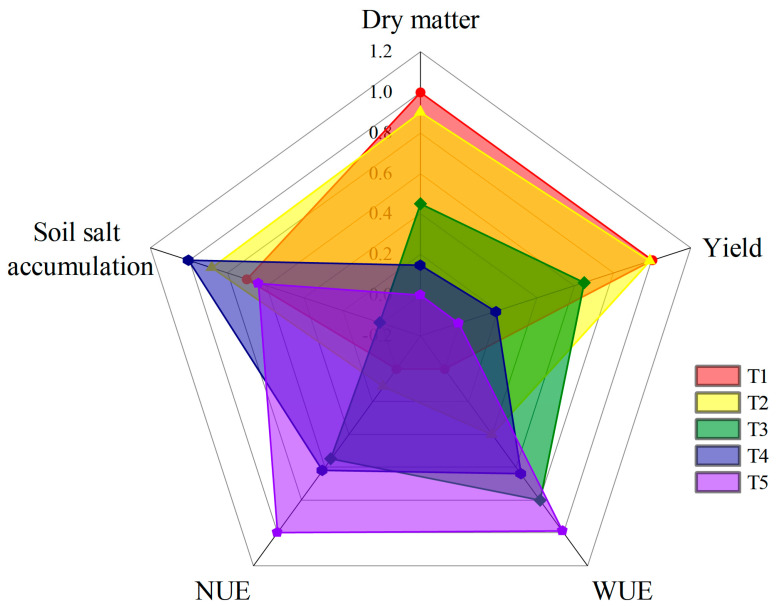
Comparison of dry matter, yield, WUE, NUE, and soil salt accumulation after the value normalized.

**Figure 8 plants-12-00791-f008:**
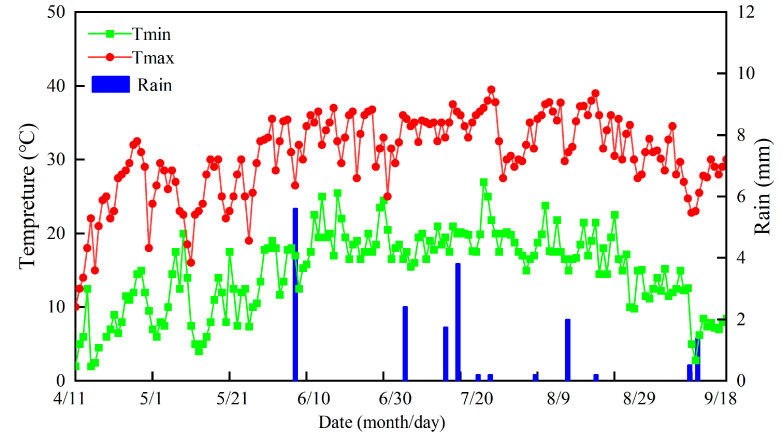
Variation of temperature and rainfall in 2018.

**Figure 9 plants-12-00791-f009:**
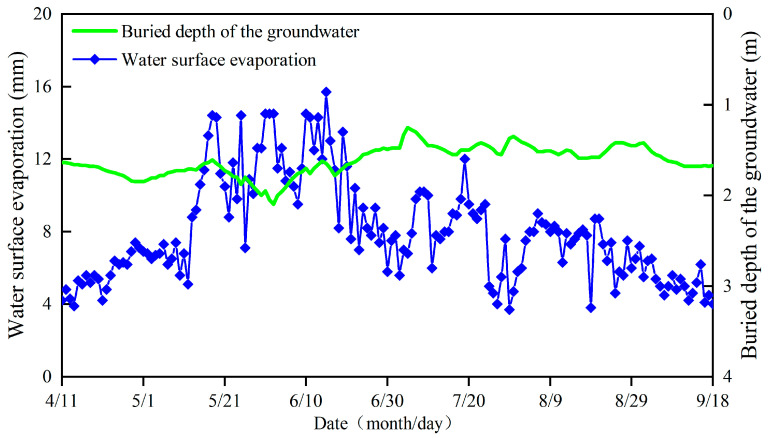
Variation of water surface evaporation and buried depth of the groundwater in 2018.

**Figure 10 plants-12-00791-f010:**
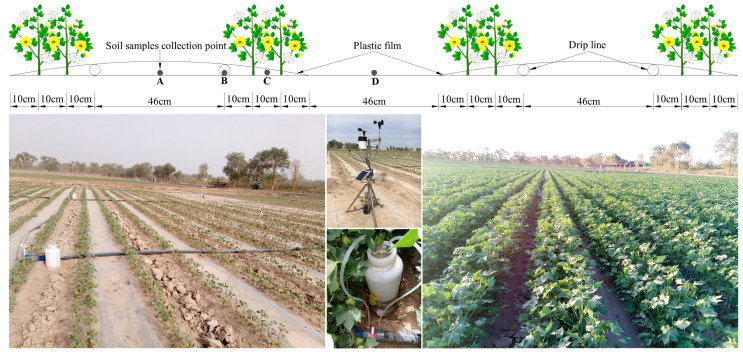
Experimental set-up and soil sampling locations.

**Table 1 plants-12-00791-t001:** Soil physical and chemical properties at the experimental site.

Depth (cm)	Soil Texture	Soil Mechanical Composition (%)			Bulk Density (g∙cm^−3^)	FC (cm^3^·cm^−3^)	Wilting Point (cm^3^·cm^−3^)	Initial Soil Water Content (cm^3^·cm^−3^)	Initial Soil Salinity (g∙kg^−1^)
		Clay	Silt	Sand					
0–10	Sandy loam	2.00	43.54	54.46	1.59	32.87%	18.60%	24.18%	0.46
10–20	Silt loam	3.30	49.30	47.41	1.44	32.86%	20.52%	21.63%	0.43
20–40	Silt loam	2.83	51.13	46.05	1.63	27.35%	13.79%	32.45%	0.46
40–60	Sandy loam	3.12	44.79	52.09	1.57	31.23%	18.60%	38.78%	0.53
60–80	Sandy	0.00	10.16	89.84	1.70	20.74%	11.82%	34.78%	0.63
80–100	Sandy	0.00	6.80	93.20	1.66	20.54%	11.70%	32.34%	0.63

**Table 2 plants-12-00791-t002:** Scheme of the mulched drip irrigation for treatments.

Treatment	Irrigation Quota and Times at Growth Stages					Total Irrigation Amount (mm)	Crop Water Consumption (mm)
	Seedling Stage	Budding Stage	Flowering Stage	Boll Setting Stage	Boll Bearing Stage		
	(11 April~5 June)	(6 June~30 June)	(1 July~25 July)	(26 July~19 August)	(20 August~20 September)		
T1	15 × 1	13 × 6	19 × 6	19 × 7	19 × 2	378	568
T2	15 × 1	21 × 3	32 × 3	32 × 4	32 × 1	334	509
T3	15 × 1	29 × 2	45 × 2	45 × 2	0	253	436
T4	15 × 1	38 × 1	58 × 1	58 × 1	58 × 1	227	412
T5	15 × 1	46 × 1	70 × 1	70 × 1	0	201	367

Note: “15 × 1” means the irrigation quota was 15 mm and the number of irrigation times wa.

## Data Availability

The data presented in this study are available on request from the corresponding author.

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
