# Peer review of "Soil Moisture Regulation under Mulched Drip Irrigation Influences the Soil Salt Distribution and Growth of Cotton in Southern Xinjiang, China"

_plants, 2023, doi:10.3390/plants12040791_

Round 1

Reviewer 1 Report

In the present paper the Authors to determine : Soil salt distribution and cotton growth under mulched drip irrigation with soil moisture regulation in southern Xinjiang, China. In the manuscript, the authors try to solve three fundamental research problems:.

-        What is the spatial distribution and temporal variation of soil moisture and soil salt;

-        They has evaluate the influence of various soil moisture lower limit regulation on plant growth, cotton yield, water and nitrogen use efficience ;

-        They had been writing to optimize the proper irrigation schedule for cotton production with soil water lower limit regulation in southern Xinjiang of China.

The authors of this work made an attempt to answer, in my opinion, successfully.

The subject of the paper fits the aims of the Journal and results could be of interest for the scientific community.

The paper is not well written.

Insights:

The abstract is incorrect, please be more specific in information.

Too long introduction - please shorten it, eg. line 49-58 information is generally known that can be omitted

M&M please provide source material for USDA classification

Discusion:

line 464-469: sentences are too long and unclear

line 480-483: sentences are too long and unclear

line 493-496: sentences are too long and unclear

line 496-500 sentences are too long and unclear

line 505-510 sentences are too long and unclear

line 522-527 sentences are too long and unclear

line 532-535 not grammatically

line 540 reference 61 is incorrect here because it concerns the root system of maize

Author Response

Dear Editors and reviewers:

Thank you for your letter and for the reviewers’ comments concerning our manuscript entitled “Soil salt distribution and cotton growth under mulched drip irrigation with soil moisture regulation in southern Xinjiang, China”(ID:plants-2162638). Those comments are all valuable and very helpful for revising and improving our papers, as well as the important guiding significance to our researches. We have studied comments carefully and have made correction which we hope meet with approval. Revised portion are marked in red in the paper. The main corrections in the paper and the responds to the reviewer’s comments are as flowing:

Response to Reviewer 1 Comments:

Point 1: The abstract is incorrect, please be more specific in information.

Response 1: We have added the year of the experiment and modified the sentence in the abstract.

Point 2: Too long introduction - please shorten it, eg. line 49-58 information is generally known that can be omitted

Response 2: According to the reviewer's suggestion, we have shorten the induction and deleted the information of line 49-58, line 94-96.

Point 3: line 464-469: sentences are too long and unclear

Response 3: The sentence ‘In the present study, after drip irrigation, the topsoil salt inside the film migrated with the wetting peak, the farther away from the drip irrigation belt of the position in the horizontal direction, the higher of the soil salinity, which followed drip line < narrow row zone < wide row zone < inter film zone, however, the law of soil water content was inverse’ was changed as ’ In this study, the salt of topsoil inside the film migrated with the wetting peak after drip irrigation. In the horizontal direction, the farther away from the drip irrigation belt, the higher of the soil salinity, which followed drip line < narrow row zone < wide row zone < inter film zone, however, the distribution of soil moisture was inverse’.

Point 4: line 480-483: sentences are too long and unclear

Response 5: The sentence ’ Many researches have proposed that soil salt accumulation in the vertical direction mainly concentrated in the surface layer, and there was little difference for the amount of soil salt accumulation in deep layers [46-48], but the threshold depth for soil salt accumulation was inconsistent’ was changed as ‘Many researches have proposed that in the vertical direction soil salt mainly accu-mulated in the surface layer rather than the deep layers [37-39]’.

Point 5: line 493-496: sentences are too long and unclear

Response 5: The sentence ‘In general, the irrigation quota and irrigation times tend to exhibit difference according to various soil moisture lower limit regulation measures, moreover, the increase of irrigation quota and irrigation times was favorable to alleviate the soil salt accumulation in the cotton field under mulched drip irrigation [42]’ was changed as ‘In the cotton field, the soil moisture regulation under mulched drip irrigation directly influenced the irrigation quota and irrigation times, another, increasing irrigation quota and irrigation times is conducive to alleviating soil salt accumulation [42]’ .

Point 6: line 496-500 sentences are too long and unclear

Response 6: The sentence ‘According to Wang et al. [15], when the soil water lower limit for irrigation was settled high, the irrigation frequency and crop water consumption increased, the horizontal distance and vertical scope of soil wetted volume decreased, while with lower soil moisture lower limit for irrigation, both the single irrigation quota and the total irrigation amount increased, which was benefit for soil salt management.’ was changed as ’ Wang et al. [15] found that the leaching effect of high irrigation lower limit on salt is lower than that of low irrigation lower limit as the horizontal distance and vertical scope of soil wetted volume is smaller.’

Point 7: line 505-510 sentences are too long and unclear

Response 7: The sentence ’ Which result was coherent with the result of Tedeschi et al. [52], who implied that the higher frequency irrigation result in the higher soil moisture and less soil salt accumulation, while lower frequency irrigation may lead to more soil salt being leached out of the crop root zone due to the higher irrigation amount. Moreover, Zhang et al. [53] indicated that the efficiency of soil salt washing with high irrigation frequency was higher than that with low irrigation frequency, which could effectively prevent salt accumulation again’ was changed as ‘High irrigation frequency could effectively prevent salt accumulation again, and the soil salt leaching efficiency with high irrigation frequency was higher than low irrigation frequency[44]. In this study, when the soil moisture lower limit was settled at 45% FC, it may lead more soil salt to be leached out of the crop root zone due to the higher single irrigation amount [45],’

Point 8: line 522-527 sentences are too long and unclear line 532-535 not grammatically

Response 8: The sentence Similarly, in this experiment, the plant height and GDM of cotton plants showed significant difference among various soil moisture lower limits, drip irrigation that triggered by soil moisture with lower limit too high (85% FC) tend to result in the overgrowth of cotton plants, while the lower soil water lower limit (45% FC) usually result in the undergrowth of cotton plants, both of which were detrimental for high cotton yield.’ was changed as ‘Similarly, in this experiment, drip irrigation that triggered by too high soil moisture lower limit (85% FC) tend to result in the overgrowth of cotton plants, while too low (45% FC) usually result in the undergrowth of cotton plants, both of which were detrimental for high cotton yield’ . The sentence ‘Which was consist with the researches of Wang et al. [14], who have noted that the cotton plant height and LAI increased with the increase of soil water lower limit, and low irrigation lower limit would produce salt stress and inhibit plant growth’ was changed as ‘Which was consist with the researches of Wang et al. [14], who have noted that the cotton plant height and LAI decreased with the decrease of soil water lower limit, as low irrigation lower limit would produce salt stress and inhibit plant growth’.

Point 9: line 540 reference 61 is incorrect here because it concerns the root system of maize

Response 9: According to the reviewer's suggestion, reference 61 was deleted.

We tried our best to improve the manuscript and made some change in the manuscript. These changes will not influence the content and framework of the paper. We appreciate for your warm work earnestly, and hope that the correction will meet with approval.

Once again, thank you very much for your comments and suggestions.

Reviewer 2 Report

Comments for authors

This is an interesting article and may be consider to publish in the journal, if author agreed to consider the following revisions:

- I do not see the year of research in the article; it is important for reviewer as well as reader

-The title of the article is complex and difficult to understand. As per the contents of the study, the title may be as: ‘Soil salt distribution and mulched based drip irrigation system influence the soil moisture and growth of cotton in southern Xinjiang, China’

- In the first line of the Abstract, you directly indicated that Water deficiency together with soil salinization have been seriously restricting the sustainable agricultural in southern Xinjiang, China for a long time. But, Soil salinity is not only problem in China, but also a common problem across the globe, particularly in coastal and arid regions. My suggestion, indicate the problem for world then China.

- Authors did not indicate the year of the observation in Abstract, it is important for it’s reality

- What do you mean ‘kg·hm-1’?

-Same to Abstract, authors should highlight the problem of salinity for Worldwide initially, then locality

-I suggest to authors for updating citations/references with last five years; since many findings have already done earlier and also available online

- In lines 96-97 and also in the whole article, better to write, ‘Thus, the experiment was conducted with five soil moisture lower limit levels, which were 85, 75, 65, 55 and 45% FC, in Korla of Xinjiang, China’; instead of ‘Thus, the experiment was conducted with five soil moisture lower limit levels, which were 85% FC, 75% FC, 65% FC, 55% FC and 45% FC, in Korla of Xinjiang, China’. Authors should follow the same for other parts of the article.

- I also suggest to authors to recheck the grammatical errors before resubmission of the revised article.

- In which year of temperature and rainfall have been in Figure 2?? Why not other meteorological parameters (such as humidity, sunshine hrs etc.)?, since these are also liked with the growth and development of plants

- Treatments should be like: T1: 5%FC, T2: 75%FC, T3: 65%FC, T4: 55%FC and T5: 45%FC; instead of ‘5%FC (T1), 75%FC (T2), 65%FC (T3), 55%FC (T4) and 45%FC (T5)’

- Try to improve the quality of all Figures

- Discussion section need to update by adding latest findings as compared to authors’ findings

Finally I recommend to publish after a minor revision

Author Response

Dear Editors and reviewers:

Thank you for your letter and for the reviewers’ comments concerning our manuscript entitled “Soil salt distribution and cotton growth under mulched drip irrigation with soil moisture regulation in southern Xinjiang, China”(ID:plants-2162638). Those comments are all valuable and very helpful for revising and improving our papers, as well as the important guiding significance to our researches. We have studied comments carefully and have made correction which we hope meet with approval. Revised portion are marked in red in the paper. The main corrections in the paper and the responds to the reviewer’s comments are as flowing:

Response to Reviewer 2 Comments:

Point 1: I do not see the year of research in the article; it is important for reviewer as well as reader

Response 1: We have added the year of research in the article, such as the abstract and Figure 2 and Figure 3.

Point 2: The title of the article is complex and difficult to understand. As per the contents of the study, the title may be as: ‘Soil salt distribution and mulched based drip irrigation system influence the soil moisture and growth of cotton in southern Xinjiang, China’

Response 2: According to the reviewer's suggestion, the title would be better changed as ‘Soil moisture regulation under mulched drip irrigation influence the soil salt distribution and growth of cotton in southern Xinjiang, China’

Point 3: In the first line of the Abstract, you directly indicated that Water deficiency together with soil salinization have been seriously restricting the sustainable agricultural in southern Xinjiang, China for a long time. But, Soil salinity is not only problem in China, but also a common problem across the globe, particularly in coastal and arid regions. My suggestion, indicate the problem for world then China.

Response 3: According to the reviewer's suggestion, we emphasized the problem for world then China in the abstract.

Point 4: Authors did not indicate the year of the observation in Abstract, it is important for it’s reality

Response 3: We have added the the year of the observation to Abstract.

Point 5: What do you mean ‘kg·hm-1’?

Response 3: We are very sorry for our incorrect writing about ‘kg·hm-2’ in abstract and changed it to ‘kg·m-3

Point 6: Same to Abstract, authors should highlight the problem of salinity for Worldwide initially, then locality

Response 3: According to the reviewer's suggestion, we emphasized the the problem for world then China in the introduction.

Point 7: I suggest to authors for updating citations/references with last five years; since many findings

have already done earlier and also available online

Response 3: Thanks to the reviewer's suggestion, I deleted some references not very correlated.

Point 8:  In lines 96-97 and also in the whole article, better to write, ‘Thus, the experiment was conducted with five soil moisture lower limit levels, which were 85, 75, 65, 55 and 45% FC, in Korla of Xinjiang, China’; instead of ‘Thus, the experiment was conducted with five soil moisture lower limit levels, which were 85% FC, 75% FC, 65% FC, 55% FC and 45% FC, in Korla of Xinjiang, China’. Authors should follow the same for other parts of the article.

Response 3: According to the reviewer's suggestion, we have replaced the sentence in lines 96-97 and other parts of the article.

Point 9: I also suggest to authors to recheck the grammatical errors before resubmission of the revised

article.

Response 3: According to the reviewer's suggestion, we modified many sentences (such as the line 464-469, line 480-483, line 493-496, line 496-500, line 505-510 and line 522-527) because of these sentences are too long or not grammatically.

Point 10: In which year of temperature and rainfall have been in Figure 2?? Why not other meteorological parameters (such as humidity, sunshine hrs etc.)?, since these are also liked with the growth and development of plants

Response 3: We have added the year of temperature and rainfall in the title of Figure 2 and Figure 3. Other meteorological parameters (such as the annual average sunshine duration for the experimental site is 2941.8 h) were introduced in 2.1. Experimental site

Point 11: Treatments should be like: T1: 5%FC, T2: 75%FC, T3: 65%FC, T4: 55%FC and T5: 45%FC; instead of ‘5%FC (T1), 75%FC (T2), 65%FC (T3), 55%FC (T4) and 45%FC (T5)’

Response 3: The sentence ‘5%FC (T1), 75%FC (T2), 65%FC (T3), 55%FC (T4) and 45%FC (T5)’ were replaced with ‘T1: 5%FC, T2: 75%FC, T3: 65%FC, T4: 55%FC and T5: 45%FC ’ in abstract.

Point 12: Try to improve the quality of all Figures

Response 3: According to the reviewer's suggestion, Figure 1 and Figure 9 were replaced with clearer pictures.

Point 13: Discussion section need to update by adding latest findings as compared to authors’ findings

Response 3: According to the reviewer's suggestion, I deleted some references not very correlated and changed some sentences.

We tried our best to improve the manuscript and made some change in the manuscript. These changes will not influence the content and framework of the paper.

We appreciate for your warm work earnestly, and hope that the correction will meet with approval.

Once again, thank you very much for your comments and suggestions.
